# Automatic Calibration of Tool Center Point for Six Degree of Freedom Robot

Chih-Jer Lin [1], Hsing-Cheng Wang [1] and Cheng-Chi Wang [2],*

[1] Graduate Institute of Automation Technology, National Taipei University of Technology, New Taipei 106, Taiwan

[2] Department of Intelligent Automation Engineering, Graduate Institute of Precision Manufacturing, National Chin-Yi University of Technology, Taichung 411, Taiwan

* Correspondence: wcc@ncut.edu.tw

**Abstract:** The traditional tool center point (TCP) calibration method requires the operator to use their experience to set the actual position of the tool center point. To address this lengthy workflow and low accuracy, while improving accuracy and efficiency for time-saving and non-contact calibration, this paper proposes an enhanced automatic TCP calibration method based on a laser displacement sensor and implemented on a cooperative robot with six degrees of freedom. During the calibration process, the robot arm will move a certain distance along the X and Y axes and collect the information when the tool passes through the laser during the process to calculate the runout of the tool, and then continue to move a certain distance along the X and Y axes for the second height calibration. After the runout angle is calculated and calibrated by triangulation, the runout calibration is completed and the third X and Y axis displacement is performed to find out the exact position of the tool on the X and Y axes. Finally, the tool is moved to a position higher than the laser, and the laser is triggered by moving downward to obtain information to complete the whole experimental process and receive the calibrated tool center position. The whole calibration method is, firstly, verified in the virtual simulation environment and then implemented on the actual cooperative robot. The results of the proposed TCP calibration method for the case of using a pin tool can achieve a positioning deviation of 0.074 and 0.125 mm for the robot moving speeds of 20 and 40 mm/s, respectively. The orientation deviation in the x-axis are 0.089 and −0.184 degrees for the robot moving speeds of 20 and 40 mm/s, respectively. The positioning repeatability of ±0.083 mm for the moving speed of 20 mm/s is lower than ±0.101 mm for the speed of 40 mm/s. It shows that lower moving speed can obtain higher accuracy and better repeatability. This result meets the requirements of TCP calibration but also achieves the purpose of being simple, economical, and time-saving, and it takes only 60 s to complete the whole calibration process.

**Keywords:** six-axis manipulator; tool center point; calibration; laser displacement sensor

## 1. Introduction

Robotic manipulators are widely used in industrial manufacturing, while the use of collaborative arms, in addition to industrial arms, is also increasing year by year. To realize Industry 4.0, automation has become an important indicator for factory transformation and the high accuracy manufacturing procedures are even more important [1–4]. According to the International Federation of Robotics [5], the number of manipulators operating in the factories is increasing year on year, moreover, the annual installations are increasing too. It can also be understood that the role of manipulators in the industry is becoming increasingly significant. With long operating hours, high repeatability precision, and low error rate, it has certainly improved the automatic production capacity and flexibility [6]; furthermore, lowering the production duty and equipment budget. In a nutshell, to maximize the advantage of the manipulator, increasing the precision is the key point. A robot arm is most

often defined as a set of rigid linkage mechanisms connected by joints. One side is attached to an external rigid surface, called the base, and the other side can be fitted with various tools, called flanges. The end effector is based on the robot's operating position, such as the center point of the vacuum suction product or where the tip of the robot torch is actually welded, which is called the operating point, also called the tool center point (TCP) [7–9]. The traditional way of TCP calibration is mostly performed by the operator, who needs to move the robot arm to reach the actual position of the tool center point to the reference station and check its accuracy by eye, and this process is repeated six to twelve times to retrieve the most accurate TCP. In recent years, many researchers proposed different methods for TCP calibration to improve the accuracy of the robot arm. Bergström [10] provided a standard idea of using a spherical probe tool and a calibration cup to calculate the TCP. In order to define the actual tool center point, he used a soft servo to move the spherical probe tool into the calibration cup, which prevents the robot system from triggering a collision alarm, while allowing the soft servo to deactivate the proportional part of the PID position control, allowing deviations from the defined program trajectory. After loading the spherical probe tool into the calibration fixture, he then reoriented the spherical probe and recorded it at least four times. After repeating this procedure several times in different directions of the tool, the final TCP would be defined. This method is one of the contact calibration methods but it cannot be implemented on any type of machining tool and is inflexible. Guo et. al. [11] proposed a constraint method for the posture of an irregular-shaped tool in this scheme. Theoretical foundations for the four-posture calibration method of the irregular-shaped tool for dual-robot-assisted ultrasonic non-destructive testing (NDT) were presented in detail. This strategy has been successfully applied in the NDT experiment of semi-enclosed composite workpieces. Experimental results show that: the calibration method can be used to obtain the correct TCP position efficiently; the TCP orientation constraint rule can ensure the extension pole of the irregular-shaped ultrasonic probe is parallel to the axis of the semi-enclosed cylindrical workpieces; and the ultrasonic transducer axis is perpendicular to the surface of the workpiece. Fares et al. [12] studied to maximize the variance of the robot's TCP value obtained by the four-point method by using the industrial robot in a set of n points generated by a random distribution and using this set of data as the input data for the sphere fitting algorithm developed in MATLAB. Moreover, the accuracy and stability of the proposed method were subsequently validated against experimental results.

　　　Machine vision is becoming increasingly important in scientific, industrial, smart manufacturing, and medical applications due to the tremendous development of PC-based languages, vision technologies, and algorithms. Erick et. al. [13] proposed a novel calibration system that uses position sensitive calibration, position sensitive detector, and camera and laser fixtures to calibrate the TCP. In order to calibrate the TCP, the laser pointer and the TCP must be located at several positions set by the user. Borrmann et al. [14] proposed a laser tracker-based calibration method for TCP. They designed a system using a laser tracker and two tool balls that can reflect the laser beam and installed this measurement tool on the TCP. The actual TCP can be obtained by rotating the robot arm, recording the information of the tool ball with the laser tracker, and finally calculating the homogeneous transformation matrix. The advantages of using laser trackers to measure the TCP are high accuracy and operator independence but the disadvantages are that they require additional equipment, are expensive, and require special environmental conditions. Zhang et al. [15] analyzed to solve the problems of poor accuracy stability and strong operational dependence in traditional TCP calibration methods and proposed a TCP calibration method for robot-assisted puncture surgery. It is more suitable and helpful for a physician. This paper designs a special binocular vision system and proposes a vision-based TCP calibration algorithm that simultaneously identifies the tool center point position (TCPP) and tool center point frame (TCPF). An accuracy test experiment proves that the designed special binocular system has a positioning accuracy of $\pm 0.05$ mm. Comparison experiments show that the proposed TCP calibration method reduces the time consumption by 82%, improves the accuracy of TCPP by 65%, and improves the accuracy of TCPF by 52% compared to the

traditional method. Liu et.al. [16] proposed a robot TCP automatic calibration algorithm based on binocular vision measurement. A target that can be recognized by the binocular vision sensor is attached to the robot TCP. The pose transformation between the vision sensor and the robot base is calculated by taking the binocular vision three-dimensional space measurement as the constraint and combining it with the multiple translational motions of the robot end tool. After several free rotations of the end tool of the robot, the TCP takes the measurement vector of the corresponding binocular vision sensor as the stroke to carry out the hypothetical parallel movement.

The main mission of the manipulator is to follow a specific trajectory and orientation use of the end effector to reach the target point. In the field of manipulator calibration, TCP calibration has rarely been studied. The purpose to calibrate TCP is to ensure precision and efficiency every time the tool changes automatically. However, inherent error often occurs when packaging or processing so the relation between the flange and TCP has to be calibrated. The traditional way to calibrate TCP is mainly by using a quick check for if the TCP is precisely targeting the reference point at the different postures and record these different postures by the manipulator controller. Yet, the process is time-consuming and labor-intensive; moreover, for non-specialized and experienced operators, the error will be magnified. Hereby, to retain high precision when the manipulator is processing, the correction of the TCP is the key factor, in addition, this research proposes a method that is based on the content mentioned above, focuses on automatic calibration technology by using a laser sensor to process runout, and offsets calibration after installing the tool; moreover, this method can achieve an easy, low time consuming, affordable price, and ultimately realize automatic operation.

## 2. Methodology

### 2.1. Design of the Experimental Structure

In this paper, TM5M-900 robot is used to verify the calibration experiments shown in Figure 1 [17,18]. This 6-axis robot is suitable for mobile assembly applications in the automated chemical and electronic industries. It is easy to program, highly customizable, has a radius range of 900 mm, and has a payload of 4 kg. The TCP, shown in Figure 1, is the working point used by the robot with a reference coordinate system attached to the robot's flange. Typically, the robot motion is programmed to define a path relative to the reference coordinate preset, which can be represented by various coordinate systems. In addition, a robot system can have multiple TCPs in it but it can only have one TCP active at a time. The robot base coordinates are attached to the robot base, and, in this study, the base coordinate system corresponds to the world coordinate system. The wrist frame is attached to the robot flange and the surface is on the robot's last axis and can mount tools. The center of the wrist frame is located in the flange center, the six axes of the robot are coinciding with the blue z-axis of the wrist frame, and the red axis and green axis represent the X and Y axes, respectively.

First, the z-axis of the tool coordinate systems is defined as the direction extending in the tool axis direction, and the TCP, which is also the origin of the tool coordinate system, is defined at this end effector (Figure 2a) [18].

From Figure 2, we can find that since the cylindrical tool is symmetrical in the X and Y axes, the rotation error along the z-axis is negligible. In this case, if the actual position and direction of the TCP, i.e., the coordinate system of the tool, is required, only the axial direction of the tool is required and then the direction of the coordinate system can be obtained by calculation. Once the direction of the coordinate system is obtained by calculation, the position of the coordinate system can be calculated by finding the end of the tool along the axial direction (Figure 2b).

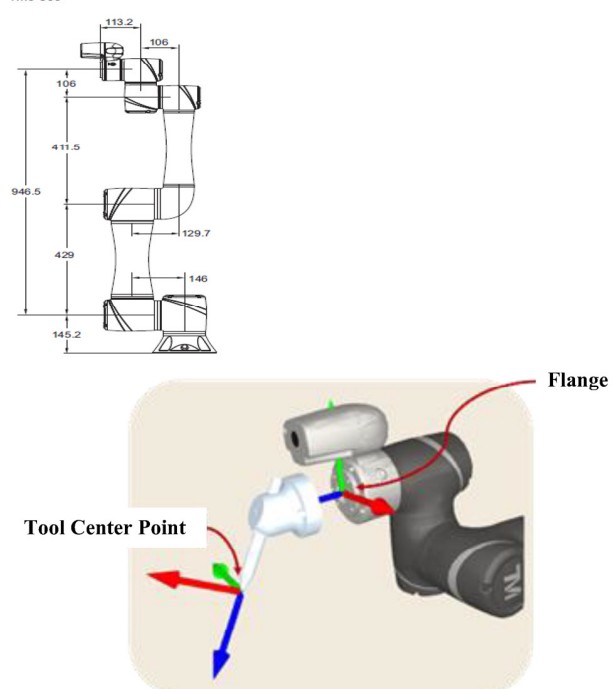

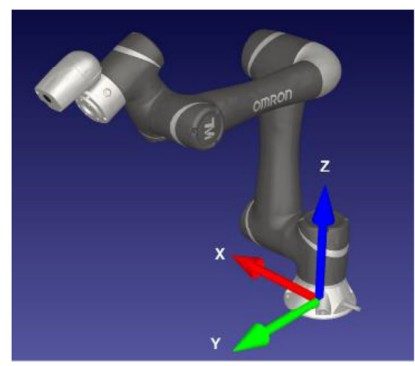

**Figure 1.** Collaborative robot TM5M-900.

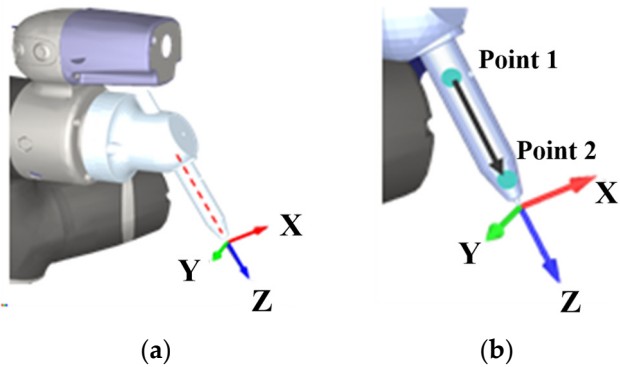

| (**a**) | (**b**) |

**Figure 2.** (**a**) Coordinate of TCP; (**b**) schematic diagram of the calculation tool axial vectors.

Secondly, we set up the laser sensor that operates by sending a Boolean value when the laser is detecting an object. The precise position of the TCP is unknown, and to ensure the TCP can be correctly detected by a laser sensor, use two laser sensors on the same plane, one in the X-axis direction and the other in the Y-axis direction, also, the calibration is executed according to this plane (Figure 3).

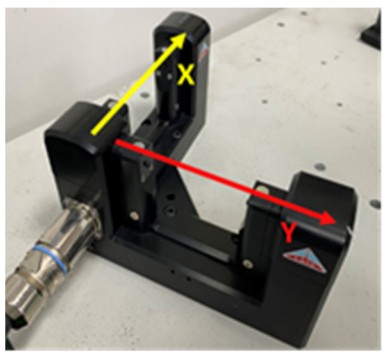

**Figure 3.** The laser sensor device.

### 2.2. Tool State Analysis and Error Modeling

Before deriving the various theoretical formulas, it is necessary to define the tool state before calibration. There are four types: Ideal case, offset, runout, and offset with runout (Figure 4a) [19].

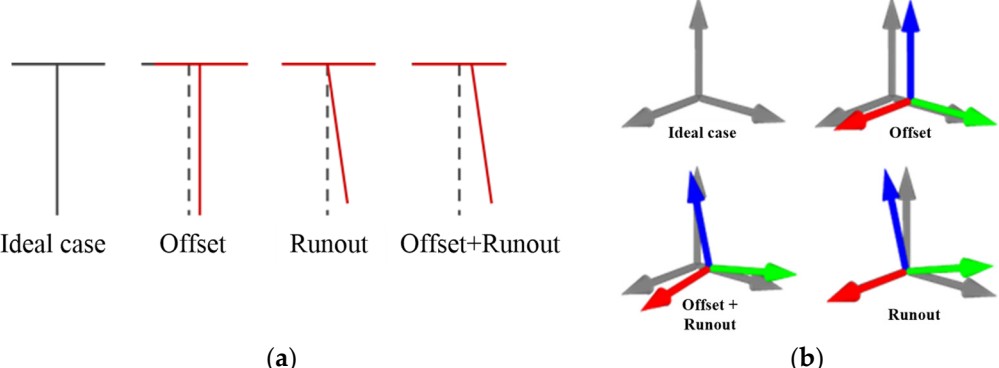

**Figure 4.** (**a**) Definition of four states of tool; (**b**) tool coordinate system.

In Figure 4, the horizontal line at the top of each state represents the flange surface, the solid black line below the horizontal line and the dark red solid line below represent the tools, and the black dashed line below each state is the ideal tool state for comparison. For the definition of each state, the tool offset represents the error of the tool's position relative to the ideal tool state, this means the tool has only the position displacement error concerning the ideal tool; the tool runout represents that the tool has only the rotation error concerning the ideal case. The tool offset and runout errors are the errors of tool position and rotation concerning the ideal case. However, in reality, when the tool is installed, the ideal tool coordinate system and the flange coordinate system usually have relative rotational deviation, so a more exact illustration of the tool condition is shown in Figure 4b.

After analyzing the possible states of the tool, the error modeling of the tool coordinate system can be performed using nonlinear equations [19], because in the kinematic model, the externally mounted tool can be considered as an extension of the robot arm; in addition, the orientation of the tool coordinate system in the robot arm base coordinate system can be expressed as a nonlinear function of the geometric parameters of the robot arm linkage, the geometric parameters of the tool, and the angular values of the joints, so the relationship between the ideal tool coordinate system and the ideal robot arm base coordinate system can be expressed as Equation (1),

$$P_{it} = f\left(\vec{q}, \vec{g_r}, \vec{g_t}\right) \tag{1}$$

where $P_{it}$ represents the measured TCP posture under the ideal robot arm base coordinate system; the vector $\vec{q}$ represents the angle value of each joint of the robot arm; vector $\vec{g_r}$ represents the ideal linkage geometric parameter of the robot arm; and the vector $\vec{g_t}$ represents the ideal tool geometric parameter.

Practically, the robot model does not anticipate the exact position and orientation of the additional mounted tools. Therefore, the difference between the actual tool pose and the robot kinematic tool pose is the geometric error of the tool installation and the link geometry error of the robot, as shown in Equation (2).

$$P_{at} = f\left(\vec{q}, \vec{g_r} + \Delta\vec{g_r}, \vec{g_t} + \Delta\vec{g_t}\right) \tag{2}$$

where $P_{at}$ represents the measured TCP posture under the actual robot arm base coordinate system; the vector $\Delta\vec{g_r}$ is the error of the geometric parameters between the ideal and the

actual robot arm connecting linkage; and the vector $\Delta \vec{g_t}$ is the error of geometric parameters between the ideal tool and the actual tool.

From Equations (1) and (2), the errors of the actual robot arm and actual tool can be derived as Equations (3) and (4).

$$\Delta P_t = P_{at} - P_{it} \tag{3}$$

$$\Delta P_t = f\left(\vec{q}, \vec{g_r} + \Delta\vec{g_r}, \vec{g_t} + \Delta\vec{g_t}\right) - f\left(\vec{q}, \vec{g_r}, \vec{g_t}\right) \tag{4}$$

where $\Delta P_t$ represents the tool coordinate and the ideal tool coordinate posture error after the actual tool linked by actual robot.

The purpose of this study is to discuss TCP calibration, so it is assumed that the robotic arm has already completed its native calibration, and the main calibration error model is shown in Equation (5).

$$\Delta P_{tool} = f\left(\vec{q}, \vec{g_{ra}}, \vec{g_t} + \Delta\vec{g_t}\right) - f\left(\vec{q}, \vec{g_{ra}}, \vec{g_t}\right) \tag{5}$$

where $\Delta P_{tool}$ represents the posture error model of the actual tool coordinate system concerning the ideal tool coordinate system after the robot has been calibrated; and the vector $\vec{g_{ra}}$ represents the linkage geometric parameters after the robot arm has been calibrated.

In the next section, the calibration theory will be derived for $\vec{g_t}$ in Equation (5), i.e., the error in the geometric parameters between the actual tool and the ideal tool. In the parameter derivation section, how to obtain the actual tool attitude will be discussed, which is mainly for the calibration of runout and offset.

### 2.3. Runout Calibration

Runout calibration can be divided into three steps. First, the tool center offset is calculated by position method and then the projection angle is derived. Finally, the rotation matrix can be solved to express the position of an object rotating in space. The position method is used to derive the tool center offset and record the position of the tool when the laser beam is triggered. As the laser sensor is triggered, the coordinates of the TCP are recorded in real-time, and four positions of the TCP are known in each moving cycle. Figure 5a shows the actual and hypothetical trajectories and Figure 5b,c show the position of the tool when it triggers the laser sensor. There are two laser beams, one is along the X-axis and another one is along the Y-axis, $I_{ideal}$ is the ideal into point, $I_{actual}$ is the actual tool into point, and P1 to P4 is the position when the laser sensor is triggered.

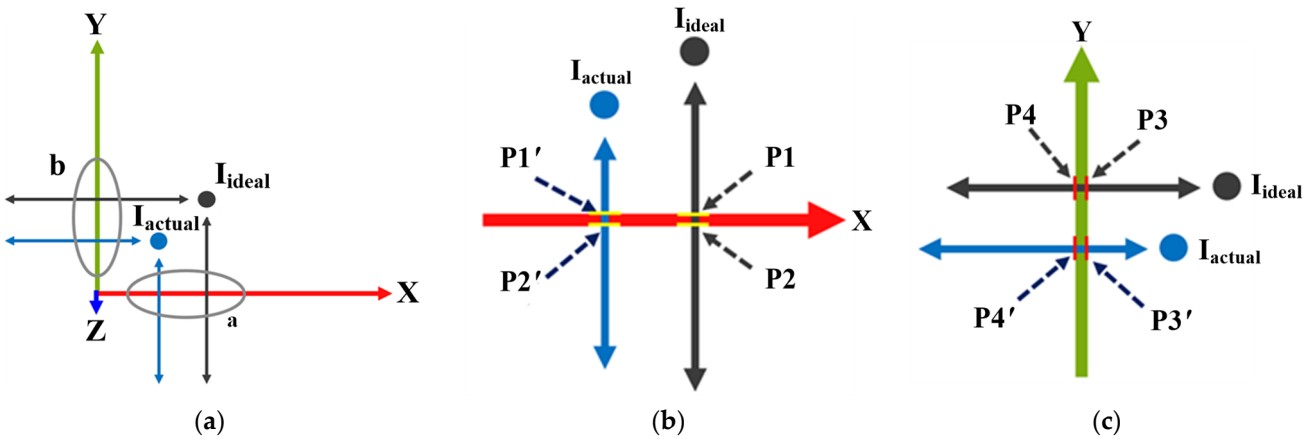

**(a)** **(b)** **(c)**

**Figure 5.** Using the four positions of the tool center offset in the same plane: (**a**) actual and hypothetical trajectories; (**b**) trigger point in X-axis; (**c**) trigger point in Y-axis.

After calculating the tool center offset, the runout angle can be obtained by performing a trigonometric calculation by the height difference. Figure 6a represents the ideal tool coordinate system ($C_p$) and the runout error of coordinate system ($C_{p'}$). When the tool

has a runout error, the base point of the tool on the flange coordinate system is defined as $(\hat{X}_p, \hat{Y}_p, \hat{Z}_p)$ and the projection angle is defined as $\theta_y$ in Y-axis. As two laser sensor units are parallel to the X and Y axes of the coordinate system, then the tool contacts the X and Y laser four times during the linear motion of the laser device. The projection angle $\theta_y$ can be found by Equation (6).

$$\theta_y = \tan^{-1}\left(\frac{\Delta x2 - \Delta x1}{z2 - z1}\right) \tag{6}$$

where $\Delta x1$ and $\Delta x2$ are the tool center offset value at plane one and plane two, respectively; $z1$ and $z2$ are the value of two planes in the $z$-axis.

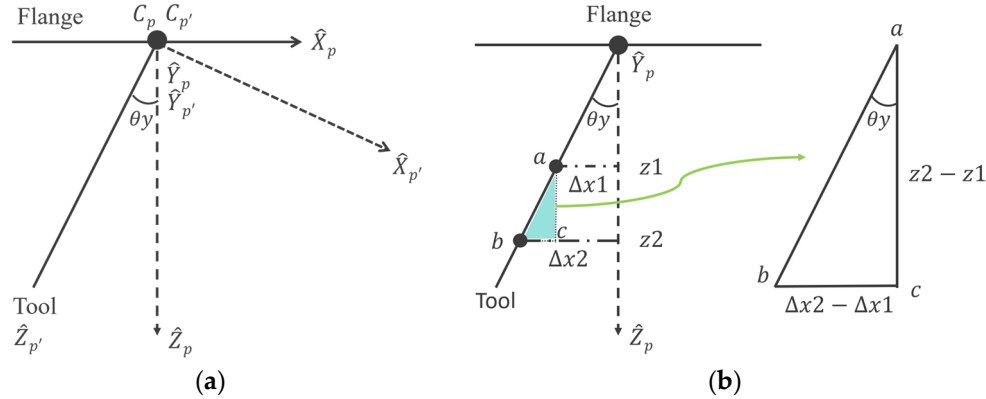

**Figure 6.** Tool projection angle: (**a**) with a relative displacement of Z-axis; (**b**) geometric relation.

In order to express the relationship of rotation matrix and projection angle, this study uses the Euler angle of the X-Y-Z rotation sequence. The first rotation is defined as an angle $\alpha$ counterclockwise around the X-axis with rotation matrix, $R_A$, and the second rotation is counterclockwise rotating with an angle $\beta$ around the Y-axis by matrix $R_B$. For the rotation of the Z-axis, it does not affect the error. The final rotation matrix ($R_d$) for the ideal tool coordinate system is shown as Equation (7):

$$R_d = \begin{bmatrix} \cos\beta & 0 & \sin\beta \\ \sin\alpha\sin\beta & \cos\alpha & -\sin\alpha\cos\beta \\ -\cos\alpha\sin\beta & \sin\alpha & \cos\alpha\cos\beta \end{bmatrix} \tag{7}$$

Assume that there is a point $P$ in the ideal tool coordinate system and it will become point $Q$ after rotating to a new coordinate system. Then, position of $Q$ can be calculated and obtained. Accordingly, the rotation matrix of the current tool ($R$) is derived by the initial tool rotation matrix ($R_0$) and the ideal tool coordinate system ($R_d$) shown in Equation (8).

$$R = R_0 \cdot R_d \tag{8}$$

### 2.4. Offset Calibration

Figure 7 shows that $O$, $O'$, and $O''$ are the ideal tool coordinate, runout tool coordinate, and runout with offset tool coordinate system, respectively. $S^c$ ($Xs$, $Ys$, $Zs$) is the tool installed station, and $P^c$ ($X_0$, $Y_0$, $Z_0$), $P^{c'}$ ($X_0'$, $Y_0'$, $Z_0'$), and $P^{c''}$ ($X_0''$, $Y_0''$, $Z_0''$) are the original points of the ideal tool coordinate, runout tool coordinate, and runout with offset tool coordinate system, respectively.

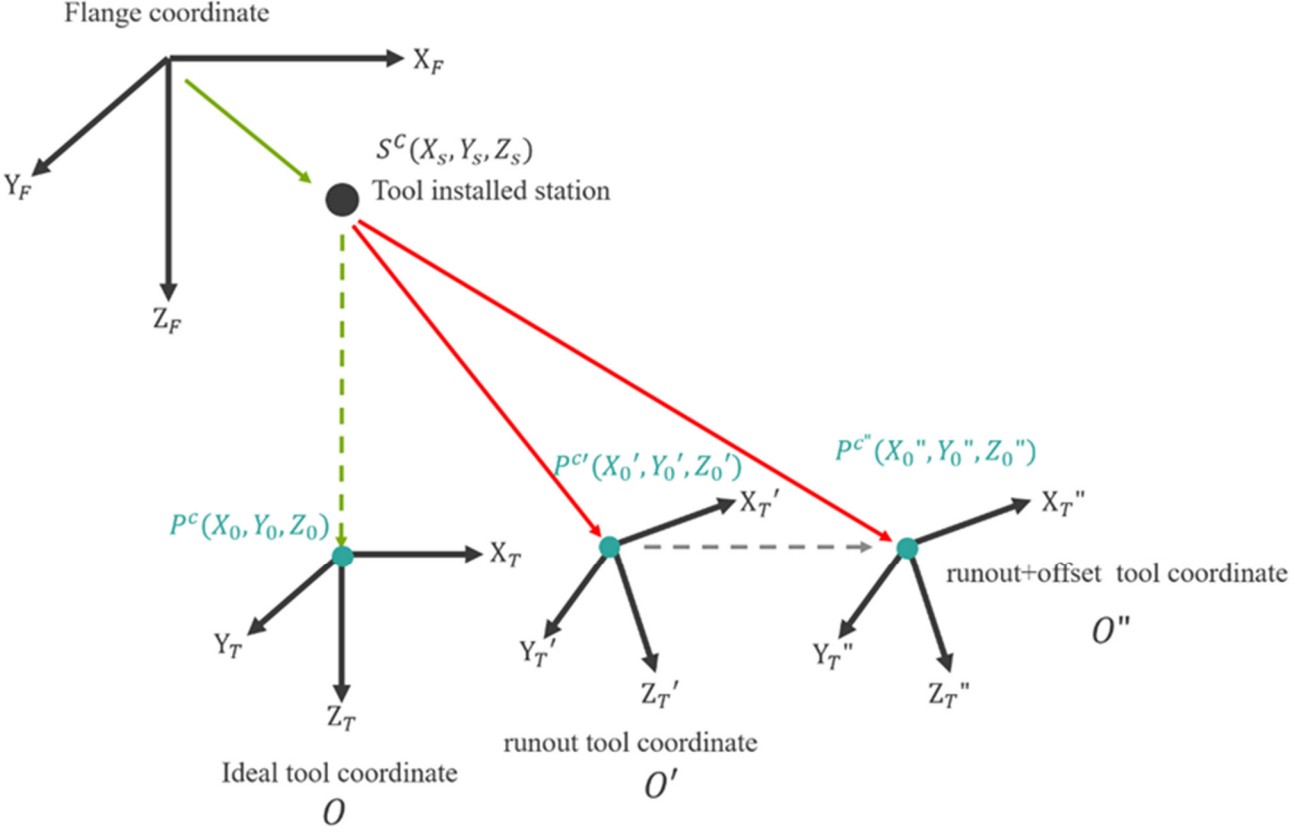

**Figure 7.** Coordinate space model.

According to the relationship between the coordinate systems, the relative equation can be derived as follows.

$$P^c = \begin{bmatrix} X_0 \\ Y_0 \\ Z_0 \end{bmatrix} = S^c + R_O \cdot Z^T = \begin{bmatrix} X_s \\ Y_s \\ Z_s \end{bmatrix} + R_O \cdot \begin{bmatrix} 0 \\ 0 \\ Z_h \end{bmatrix} \tag{9}$$

$$P^{c'} = \begin{bmatrix} X_0' \\ Y_0' \\ Z_0' \end{bmatrix} = S^c + R_O \cdot (R_R \cdot Z^T) = \begin{bmatrix} X_s \\ Y_s \\ Z_s \end{bmatrix} + R_O \cdot R_R \cdot \begin{bmatrix} 0 \\ 0 \\ Z_h \end{bmatrix} \tag{10}$$

$$
\begin{aligned}
P^{c''} = \begin{bmatrix} X_0'' \\ Y_0'' \\ Z_0'' \end{bmatrix} &= P^{c'} + R_O \cdot R_R \cdot \delta R^{T'} = \begin{bmatrix} X_0' \\ Y_0' \\ Z_0' \end{bmatrix} + R_O \cdot R_R \cdot \begin{bmatrix} \delta X \\ \delta Y \\ \delta Z \end{bmatrix} = P^c + R_O \cdot \left( R_R \cdot \delta R^{T'} - Z^T \right) \\
&= \begin{bmatrix} X_0 \\ Y_0 \\ Z_0' \end{bmatrix} + R_O \begin{bmatrix} \delta X \cos \beta + (Z_h + \delta Z) \sin \beta \\ \delta X \sin \alpha \sin \beta + \delta Y \cos \alpha - (Z_h + \delta Z) \sin \alpha \cos \beta \\ -\delta X \cos \alpha \sin \beta + \delta Y \sin \alpha + (Z_h + \delta Z) \cos \alpha \cos \beta - Z_h \end{bmatrix}
\end{aligned}
\tag{11}
$$

where $Z_h$ is the distance between $P^c$ and $S^c$ and expressed as a spatial vector $Z^T$.

The calibration process is shown in Figure 8. In the runout calibration, the actual tool information is obtained after two planes of motion and the tool center offset parameters can be obtained, which are then used to calculate the runout angle and rotation matrix. The first preliminary offset calibration is also performed. After completing the runout calibration, a second preliminary offset calibration is executed using the tool information obtained from the third plane motion to accurately calculate the X and Y axis offset parameters based on the tool coordinate system. The last process is the final offset calibration. The tool moves over the laser beam, triggering the laser sensor to go vertically down and find the Z-axis parameters of the TCP according to the tool coordinate system.

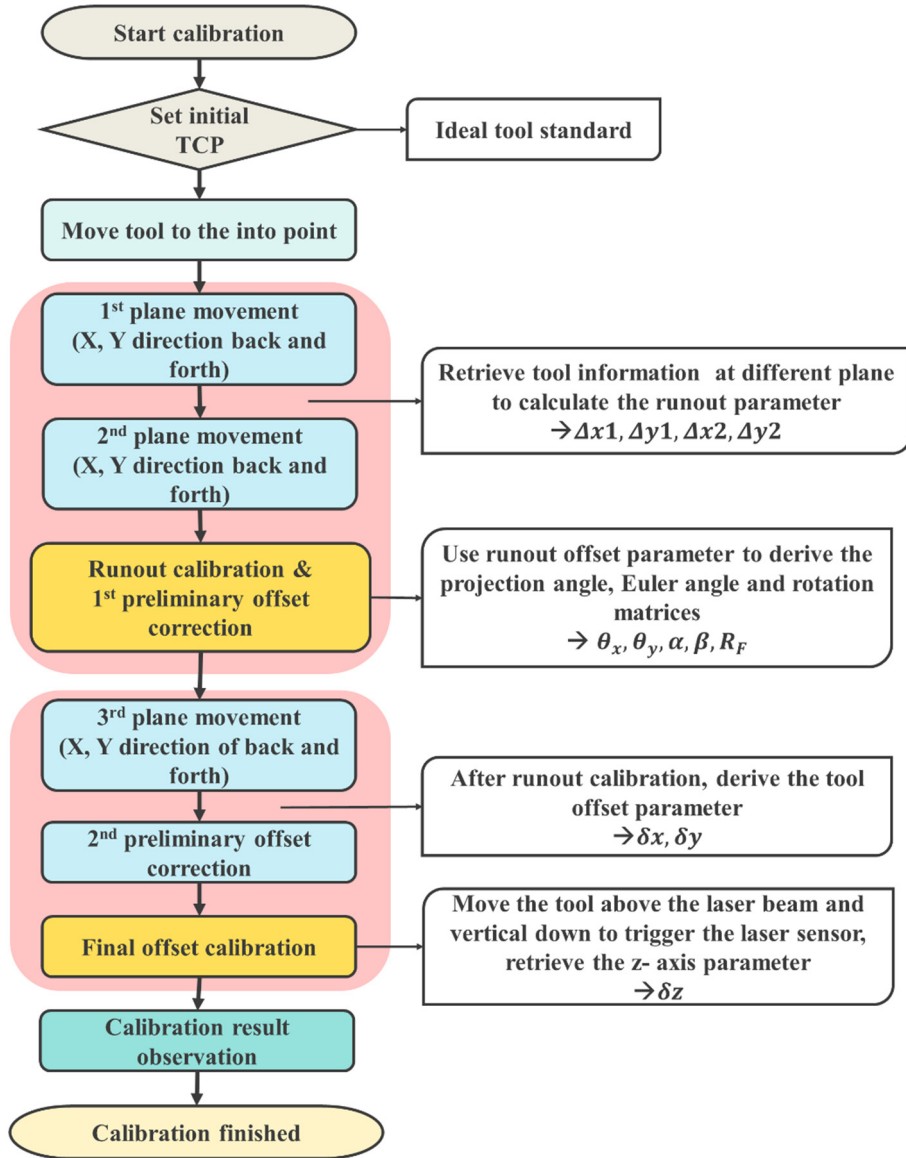

**Figure 8.** Flow chart of calibration process.

In Figure 8, the first preliminary offset calibration is performed after the 2nd plane movement is completed. The first preliminary offset calibration, which estimates the tool height $Z_h$, can be performed when the tool only has an offset. The second preliminary offset calibration is performed by deriving the tool offsets $\delta X$ and $\delta Y$ from the tool center offset equation. Figure 9 represents how the tool height $Z_h$ is calculated.

Looking in from the positive direction of the Y-axis of the coordinates of the tool mounting point, where $O$ is the tool mounting station and the origin of the coordinate system, the coordinates of $p^o$ are $(0, 0, Z_h)$ and the coordinates of $p'^o$ are $(x', y', z')$. Let the first plane of calibrated motion be $M1$ and the second plane be $M2$, where $\overline{P^oO}$ intersects with plane $M1$ at $M1_s(0, 0, z1)$ and with plane $M2$ at $M2_s(0, 0, z2)$, $\overline{P'^oO}$ intersects the plane $M1$ at $M1_s(0, 0, z1)$, $\overline{P'^oO}$ intersects the plane $M2$ at $M2_s'(\Delta x2, \Delta y2, z2)$, and the parameters $\Delta x1$, $\Delta x2$, $\Delta y1$, $\Delta y1$ have been obtained from Equation (6). Let the height from plane one to $p^o$, labeled $\overline{P^oM1_s}$ be $\Delta H$, and the height between plane one and plane two, labeled $\overline{M1_sM2_s}$, be $\Delta h$. After defining the above information, the following Equations (12) and (13) can be obtained.

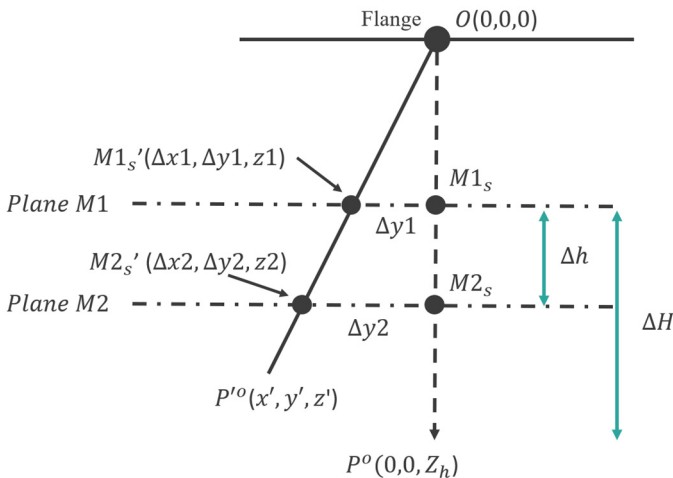

**Figure 9.** The tool height derivation chart of the tool runout coordinate system.

$$Z_h = Z1 + \Delta H \tag{12}$$

$$Z2 = Z1 + \Delta h \tag{13}$$

where $Z_h$ is the tool height that needs to be derived; $\Delta H$ and $\Delta h$ is the parameter set by the user. Use space linear proportion and the center offset parameter, which is $\Delta x1$, $\Delta x2$, $\Delta y1$, $\Delta y1$. Then, through relational substitution $Z1$ and $Z2$ can be solved.

To express the spatial linear scale, the coordinates of at least two points in space must be known. Herby, assume there are two points, A $(x1, y1, z1)$ and B $(x2, y2, z2)$, the line $\eta$ between these two points can be expressed in the spatial linear scale as Equation (14).

$$\eta \; : \; \frac{x - x1}{x2} = \frac{y - y1}{y2} = \frac{z - z1}{z2} \tag{14}$$

Similarly, since $\Delta x1$, $\Delta x2$, $\Delta y1$, $\Delta y1$, $z1$, $z2$ can construct two points $M1_s{'}$ and $M2_s{'}$, the linear $H$ is expressed by the spatial linear scale,

$$H \; : \; \frac{x - \Delta x1}{\Delta x2} = \frac{y - \Delta y1}{\Delta y2} = \frac{z - z1}{\Delta z2} \tag{15}$$

In this case, since the line $H$ passes through the tool mounting point, after substituting Equation (15) and combining Equation (13) with Equation (12), then we can obtain $Z_h$ as Equation (16).

$$Z_h = \frac{\Delta h \Delta x1}{\Delta x2 - \Delta x1} + \Delta H = \frac{\Delta h \Delta y1}{\Delta 2 - \Delta y1} + \Delta H \tag{16}$$

After the above calculation, it seems that the tool height $Z_h$ has been solved; however, a tool with offset error will result in a denominator equal to zero, while a tool with offset and offset error will result in a line $H$ not passing through the tool mounting point $O$. Therefore, for runout error or runout plus offset error, the original tool height will be used directly as the tool length $Z_h$.

The first preliminary offset calibration only roughly calculated the tool height, the runout plus offset error of the coordinates $\delta R^{T'}$ still can not be solved, so in the first preliminary offset calibration, use $Z_h$ generation back to Equation (11) to obtain the tool center point that is the current tool coordinate origin, set to $P^{c''}{}_{temp1}$, and the coordinates will be returned to the robot arm controller; the first preliminary offset calibration is completed.

### 2.5. Calibration Environment Design

A virtual environment was used to verify the hypothesis algorithm to then be implemented in the real manipulator and laser sensor device. RoboDK software and Python

were applied for the virtual environment and coding. In the real environment, the TM5-900 manipulator was used with LATC laser sensor, and the code was based on C# to complete the experiment. Figure 10 shows the actual experiment structure setting. The specification of TM5M-900 and LATC laser displacement sensor are shown in Tables 1 and 2, respectively.

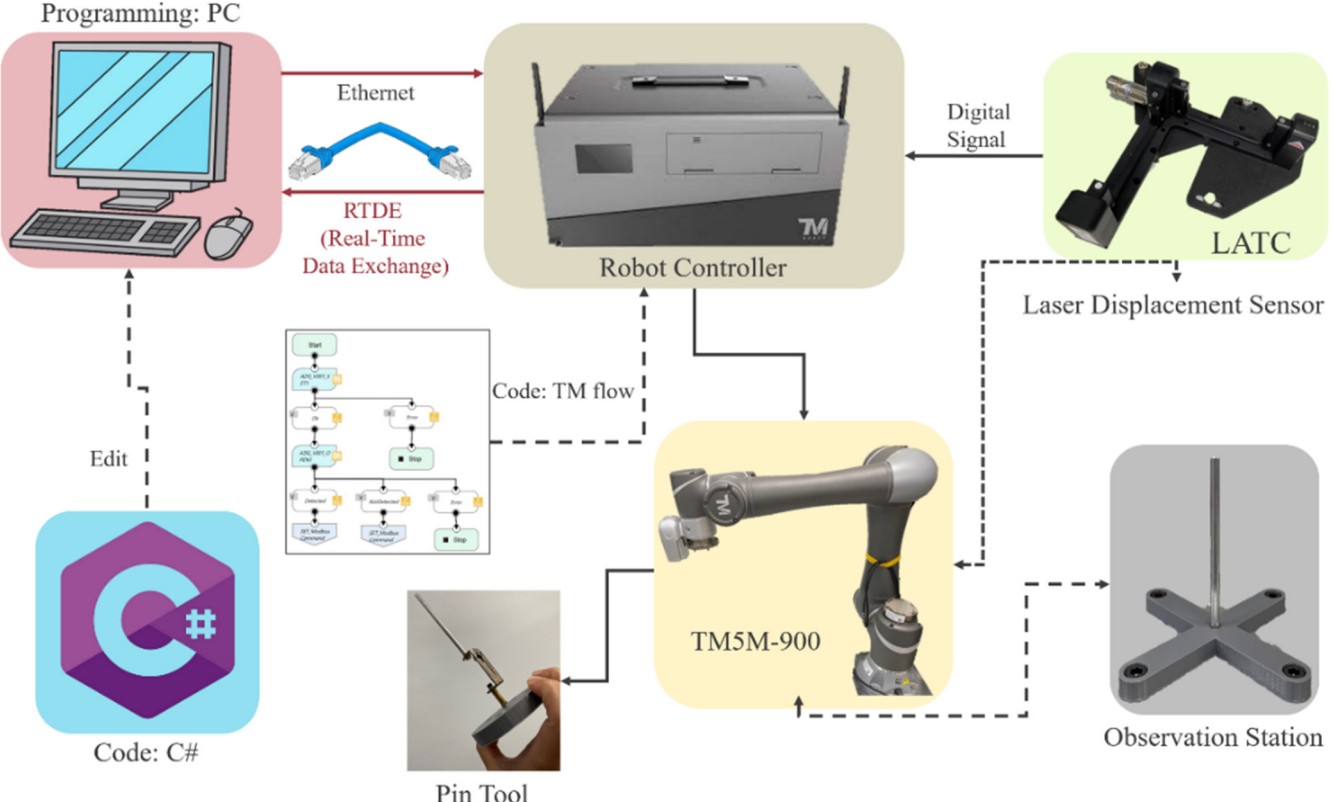

**Figure 10.** Experimental environment system structure.

**Table 1.** TM5M-900 specification.

| TM5M-900 Specification | | | |
|---|---|---|---|
| Weight | 22.6 kg | Typical speed | 1.4 m/s |
| Max Payload | 4 kg | Operating Temperature | 0 to 50 °C |
| Reach | 900 mm | Collaboration | Yes |
| Repeatability | ±0.05 mm | DOF | 6 |

**Table 2.** LATC laser displacement sensor LTC120120 specification.

| LATC Laser Displacement Sensor LTC120120 | | | |
|---|---|---|---|
| Supply Voltage | 24Vdc | Tool Size | $\varnothing$ = 0.5~100 mm |
| Supply Current | 0.2 A | Laser Type | Class 2, Red light Wavelength = 650 nm |
| Working Range | 120 × 120 mm | Working Temperature | 5 to 55 °C |
| Repeatability | <1 μm | Waterproof | IPX8 |

When the tool enters the sensing range of the sensor and triggers the sensor, it sends a digital signal to the robot controller and returns to the computer side. Figure 11a shows the lab environment setup and Figure 11b shows the standard calibration pin tool.

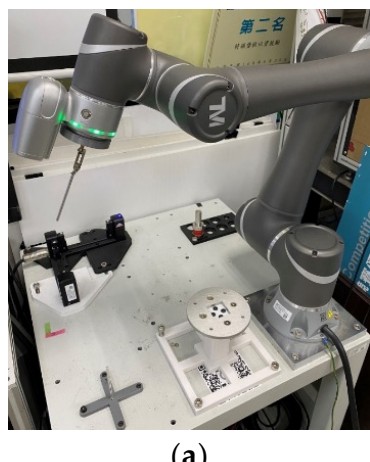
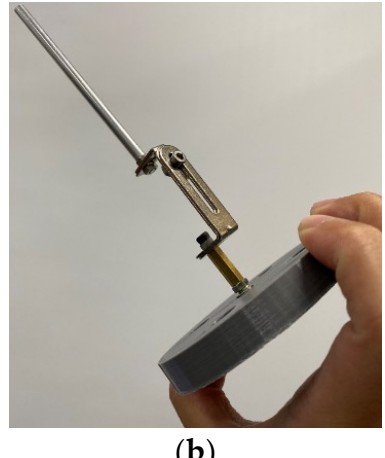

(**a**)                          (**b**)

**Figure 11.** Experiment setup: (**a**) with a TM5-900 manipulator; and (**b**) standard calibration pin tool.

## 3. Results and Discussion

In each experiment, in addition to the calibrated tool center position data, the calibration error and stability of each set of calibration data are also calculated. The calculation of calibration error and stability is based on ISO 9283, the performance standard for industrial robotic arms and related test methods, which can be followed to calculate the absolute accuracy of posture and the accuracy of posture repetition. The purpose of the simulation is to analyze the feasibility of automatic TCP correction with different tools and different arm movement speeds.

### 3.1. Error-Free Stability Simulation with the Position Method

The first simulation uses different moving speeds, an error-free tool, and position methods to handle automatic tool center point calibration, and the whole process will be performed 30 times consecutively. The robot moving speeds are set to 20 mm/s and 50 mm/s with an 8 mm radius of pencil tool and a 20 mm radius of cylinder tool, respectively. Tables 3 and 4 show the simulation results of error-free stability by the pencil tool and cylinder tool, respectively.

**Table 3.** Simulation data of error-free stability using a pencil tool.

|  | Robot moving speed | $X_d$ (mm) | $Y_d$ (mm) | $Z_d$ (mm) | $\theta x_d$ (°) | $\theta y_d$ (°) |
|---|---|---|---|---|---|---|
| Default Value | 20 mm/s | 0 | −82.7 | 101.73 | 0 | −60 |
|  | 50 mm/s | 0 | −82.7 | 101.73 | 0 | −60 |
|  | Robot moving speed | $X_m$ (mm) | $Y_m$ (mm) | $Z_m$ (mm) | $\theta x_m$ (°) | $\theta y_m$ (°) |
| Mean Value | 20 mm/s | 0.139 | −82.799 | 101.559 | 0 | −60 |
|  | 50 mm/s | 0.218 | −81.278 | 101.581 | 0.506 | −60.525 |
|  | Robot moving speed | $P_a$ (mm) |  | $\theta x_a$ (°) |  | $\theta y_a$ (°) |
| Acuracy | 20 mm/s | 0.241 |  | 0 |  | 0 |
|  | 50 mm/s | 1.841 |  | 0.506 |  | −0.525 |
|  | Robot moving speed | $P_r$ (mm) |  | $\theta x_r$ (°) |  | $\theta y_r$ (°) |
| Repeatability | 20 mm/s | ±0 |  | ±0 |  | ±0 |
|  | 50 mm/s | ±3.656 |  | ±4.645 |  | ±4.789 |

**Table 4.** Simulation data of error-free stability using a cylinder tool.

| | Robot moving speed | $X_d$ (mm) | $Y_d$ (mm) | $Z_d$ (mm) | $\theta x_d$ (°) | $\theta y_d$ (°) |
|---|---|---|---|---|---|---|
| Default Value | 20 mm/s | 0 | 0 | 110 | 0 | 0 |
| | 50 mm/s | 0 | 0 | 110 | 0 | 0 |
| | Robot moving speed | $X_m$ (mm) | $Y_m$ (mm) | $Z_m$ (mm) | $\theta x_m$ (°) | $\theta y_m$ (°) |
| Mean Value | 20 mm/s | −0.450 | −0.295 | 110.43 | 0 | −0.667 |
| | 50 mm/s | −0.301 | 0.185 | 110.46 | −0.242 | −0.32 |
| | Robot moving speed | $P_a$ (mm) | | $\theta x_a$ (°) | | $\theta y_a$ (°) |
| Acuracy | 20 mm/s | 0.687 | | 0 | | −0.667 |
| | 50 mm/s | 0.579 | | −0.242 | | −0.32 |
| | Robot moving speed | $P_r$ (mm) | | $\theta x_r$ (°) | | $\theta y_r$ (°) |
| Repeatability | 20 mm/s | ±0.382 | | ±0 | | ±0 |
| | 50 mm/s | ±3.809 | | ±5.017 | | ±4.257 |

It is found that the accuracy and repeatability under 20 mm/s are better than the results of 50 mm/s. The reason is when using a high-speed tool through the laser sensor, the sensor cannot be responded immediately and the error will be increased. In addition, for the RoboDK virtual environment, the triggering of the laser sensor is interfered by the 3D model. The results of 20 mm/s and 50 mm/s are quite different and there is almost no error in the case of speed 20 mm/s or the error is fixed; however, the results for speed 50 mm/s are unstable, as shown in Figure 12.

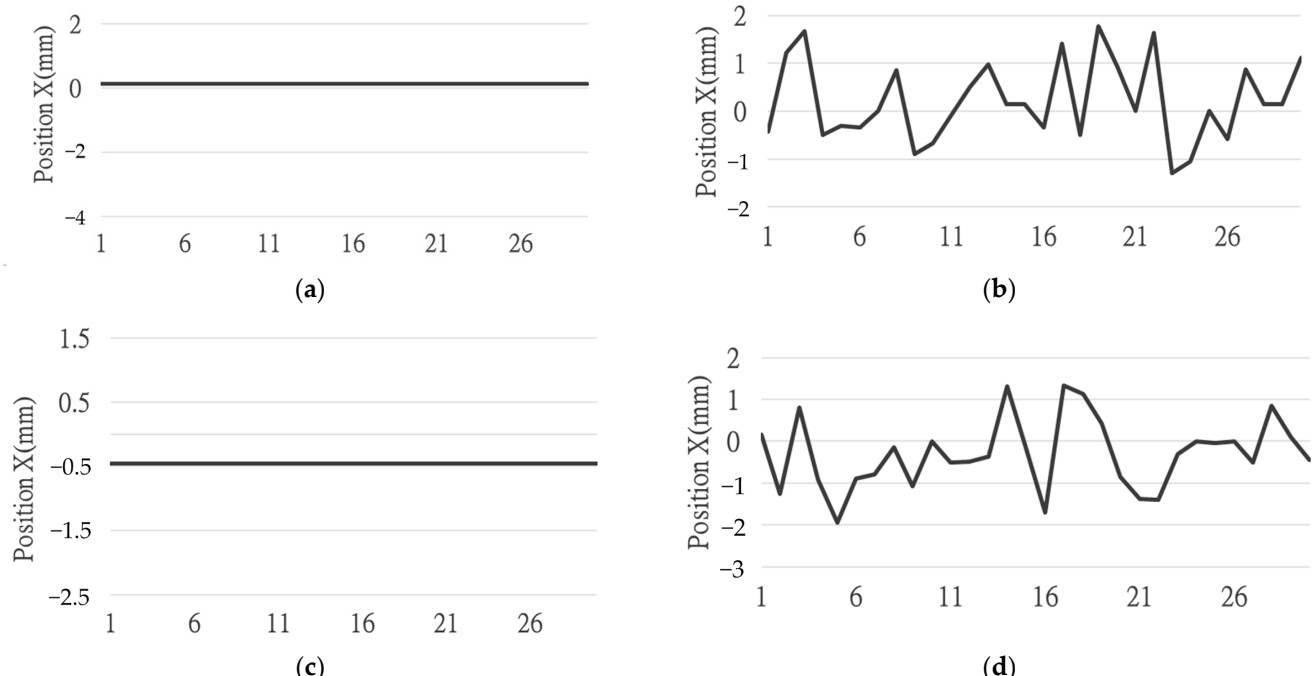

**Figure 12.** Calibration data of X position: (**a**) pencil tool at 20 mm/s; (**b**) pencil tool at 50 mm/s; (**c**) cylinder tool at 20 mm/s; (**d**) cylinder tool at 50 mm/s.

By using a pencil tool, the results of the proposed TCP calibration method can achieve a positioning deviation of 0.241 and 1.841 mm for the robot moving speeds of 20 and 50 mm/s, respectively. The orientation deviation ($\theta x_a$) was 0 and 0.506 degrees for the moving speed of 20 and 50 mm/s, respectively. The positioning repeatability was ±0.083 mm and the

orientation repeatability ($\theta x_a$) was ±4.645 for a speed of 50 mm/s. It reveals that lower robot moving speed can obtain higher accuracy and better repeatability, as shown in Table 3.

### 3.2. Four-Quadrant Calibration Simulation with the Position Method

In the four-quadrant simulation section, the experiments were conducted by changing the geometric position without changing the tool coordinates. Table 5 shows the four cases of tool center point geometry position relative to the tool coordinate error setting, i.e., four quadrants.

**Table 5.** Parameter settings for different tools geometry position in four different quadrants.

| Tool | Speed (mm/s) | Quadrant | x (mm) | y (mm) | z (mm) | $R_x$ (°) | $R_y$ (°) |
|------|--------------|----------|--------|--------|--------|-----------|-----------|
| Pencil | 20 and 50 | 1st | −1 | 1 | 0 | −5 | −5 |
|        |           | 2nd | −1 | −1 | 0 | 5 | −5 |
|        |           | 3rd | 1 | 1 | 0 | −5 | 5 |
|        |           | 4th | 1 | −1 | 0 | 5 | 5 |
| Cylinder | 20 and 50 | 1st | 1 | 1 | 0 | −5 | 5 |
|          |           | 2nd | −1 | 1 | 0 | −5 | −5 |
|          |           | 3rd | −1 | −1 | 0 | 5 | −5 |
|          |           | 4th | 1 | −1 | 0 | 5 | 5 |
|          |           | 1st | 1 | 1 | 0 | −10 | 10 |
|          |           | 2nd | −1 | 1 | 0 | −10 | −10 |
|          |           | 3rd | −1 | −1 | 0 | 10 | −10 |
|          |           | 4th | 1 | −1 | 0 | 10 | 10 |

During the experiment, the calibration for one quadrant was performed 30 times in succession, each time restoring the geometric position of the tool coordinate system and the tool center point to the pre-calibration state. The calibration accuracy and repeatability of calibration are shown in Tables 6 and 7.

**Table 6.** Four-quadrant calibration simulation results using a 5° runout (speed at 20 mm/s).

| Tool | Quadrant | | 1st | 2nd | 3rd | 4th |
|------|----------|---|-----|-----|-----|-----|
| Pencil | Accuracy | $P_a$ (mm) | 0.239 | 0.689 | 0.616 | 0.571 |
|        |          | $\theta x_a$ (°) | −0.044 | 1.716 | −1.718 | −0.405 |
|        |          | $\theta y_a$ (°) | −0.717 | 0.341 | 0.37 | −1.472 |
|        | Repeatability | $P_r$ (mm) | ±0 | ±0 | ±0 | ±0 |
|        |          | $\theta x_r$ (°) | ±0 | ±0 | ±0 | ±0 |
|        |          | $\theta y_r$ (°) | ±0 | ±0 | ±0 | ±0 |
| Cylinder | Accuracy | $P_a$ (mm) | 0.852 | 0.578 | 0.578 | 0.481 |
|          |          | $\theta x_a$ (°) | −0.705 | −0.704 | 0.704 | 0.705 |
|          |          | $\theta y_a$ (°) | −0.067 | 0.164 | 0.164 | −0.067 |
|          | Repeatability | $P_s$ (mm) | ±0 | ±0 | ±0 | ±0 |
|          |          | $\theta x_r$ (°) | ±0 | ±0 | ±0 | ±0 |
|          |          | $\theta y_r$ (°) | ±0 | ±0 | ±0 | ±0 |

**Table 7.** Four-quadrant calibration simulation results using a 5° runout (speed at 50 mm/s).

| Tool | Quadrant | | 1st | 2nd | 3rd | 4th |
|---|---|---|---|---|---|---|
| Pencil | Accuracy | $P_a$ (mm) | 0.239 | 0.689 | 0.616 | 0.571 |
| | | $\theta x_a$ (°) | −0.044 | 1.716 | −1.718 | −0.405 |
| | | $\theta y_a$ (°) | −0.717 | 0.341 | 0.37 | −1.472 |
| | Repeatability | $P_r$ (mm) | ±5.233 | ±4.710 | ±5.209 | ±5.344 |
| | | $\theta x_r$ (°) | ±5.869 | ±6.835 | ±6.870 | ±4.139 |
| | | $\theta y_r$ (°) | ±5.005 | ±4.895 | ±5.254 | ±5.335 |
| Cylinder | Accuracy | $P_a$ (mm) | 0.786 | 0.568 | 0.509 | 0.803 |
| | | $\theta x_a$ (°) | 0.016 | −0.511 | 0.602 | 0.605 |
| | | $\theta y_a$ (°) | −0.652 | 0.276 | 0.166 | −0.447 |
| | Repeatability | $P_s$ (mm) | ±3.785 | ±3.309 | ±3.077 | ±2.901 |
| | | $\theta x_r$ (°) | ±5.223 | ±4.000 | ±4.391 | ±3.709 |
| | | $\theta y_r$ (°) | ±4.449 | ±3.036 | ±3.966 | ±3.956 |

The four-quadrant calibration simulation results seem to be highly inaccurate and it is obvious that some of the errors are caused by the 3D model interfering at different quadrants. The error at the second and third quadrant are higher than the first and fourth quadrants; in addition, the misalignment are almost the same in each quadrant. Due to the laser sensor delay, the results at speed 20 mm/s can reach the standard so that this calibration method still achieves the purpose of TCP calibration shown in Table 8, so it can continue to be tested in the actual experiment.

**Table 8.** Comparison results of four-quadrant calibration simulation.

| Conditions | Accuracy | | | Repeatability | | |
|---|---|---|---|---|---|---|
| Parameters | $P_a$ (mm) | $\theta X_a$ (°) | $\theta Y_a$ (°) | $P_r$ (mm) | $\theta X_r$ (°) | $\theta Y_r$ (°) |
| 5° pencil 20 mm/s | 0.52875 | −0.11275 | −0.3695 | ±0 | ±0 | ±0 |
| 5° pencil 50 mm/s | 2.17225 | −0.04575 | 0.2995 | 5.124 | 5.92825 | 5.12225 |
| 5° cylinder 20 mm/s | 0.62225 | 0 | 0.0485 | ±0 | ±0 | ±0 |
| 5° cylinder 50 mm/s | 0.6665 | 0.178 | −0.16425 | 3.268 | 4.33075 | 3.85175 |
| 10° cylinder 20 mm/s | 0.9085 | 0 | −0.6185 | ±0 | ±0 | ±0 |
| 10° cylinder 50 mm/s | 1.10475 | 0.22225 | −0.4755 | 2.77925 | 3.59825 | 3.5135 |

### 3.3. Experiment of Error-Free Stability

In the actual error-free stability experiment, a standard of calibration tools with a processing error of ±0.05 mm are used, and the whole process is performed 30 times consecutively. After calibration, the tool is moved to the observation station to check the result. The moving speed of robot is set to be 20 mm/s and 40 mm/s, respectively.

The data of the error-free calibration experiment by the position method at different speeds are shown in Table 9. It can be seen that the misalignment at a speed of 40 mm/s is slightly larger than that at a speed of 20 mm/s. Accordingly, all results are to be considered with a positioning accuracy of ±0.05 mm for the robot arm with moving speed of 20 mm/s.

**Table 9.** Experimental result of error-free stability using a pin tool at different moving speeds.

| Default Value | | | Mean Value | | | Accuracy | | | Repeatability | | |
|---|---|---|---|---|---|---|---|---|---|---|---|
| **Speed (mm/s)** | **20** | **40** | **Speed (mm/s)** | **20** | **40** | **Speed (mm/s)** | **20** | **40** | **Speed (mm/s)** | **20** | **40** |
| $X_d$ (mm) | 0.07 | 0.644 | $X_m$ (mm) | 0.0219 | 0.654 | $P_a$ (mm) | 0.074 | 0.125 | $P_r$ (mm) | ±0.083 | ±0.101 |
| $Y_d$ (mm) | 468.653 | 468.391 | $Y_m$ (mm) | 468.70 | 468.358 | $\theta X_a$ (°) | 0.089 | −0.184 | $\theta X_r$ (°) | ±0.207 | ±0.257 |
| $Z_d$ (mm) | 84.403 | 84.349 | $Z_m$ (mm) | 84.434 | 84.228 | $\theta Y_a$ (°) | 0.182 | −0.128 | $\theta Y_r$ (°) | ±0.173 | ±0.216 |
| $\theta X_d$ (°) | 179.647 | 179.935 | $\theta X_m$ (°) | 179.736 | 179.750 | | | | | | |
| $\theta Y_d$ (°) | −0.804 | −0.312 | $\theta Y_m$ (°) | −0.621 | −0.44 | | | | | | |

By using a pin tool, there was a positioning deviation of 0.074 and 0.125 mm for the robot moving speeds of 20 and 40 mm/s, respectively. The orientation deviation ($\theta x_a$) was 0.089 and −0.184 degrees for the moving speeds of 20 and 40 mm/s, respectively. The positioning repeatability was ±0.083 mm and ±0.101 mm for the moving speeds of 20 mm/s and 40 mm/s, respectively; and the orientation repeatability ($\theta x_a$) was ±0.207 and ±0.257 for the speeds of 20 mm/s and 40 mm/s, respectively. It shows that lower moving speed can achieve higher accuracy and better repeatability, as shown in Table 9.

*3.4. Four-Quadrant Calibration Experiment*

In the four-quadrant experiment, the tool was used in the same way as the error-free experiment. Before the experiment starts, the tool is rotated to the four-quadrant position for the experiment, and then the calibration procedure is started and calibrated 30 times continuously. After the calibration is finished, the tool is moved to the observation station and the results are checked. Due to the offset of the rotating tool, the actual TCP and tool position are difficult to measure, so the stability results of the calibration are only considered. The results of the first quadrant calibration experiment with the position method at a speed of 20 mm/s are shown in Figure 13, and the experimental results have been compiled in Table 10. It can be seen that the positioning method used in the actual experiment is much more accurate than that in the simulation environment but the error of the results in the four quadrants is slightly higher than that in the error-free experiment but the positioning repeatability can reach 0.12 mm and the positioning repeatability can reach less than 0.14°.

**Table 10.** Results of the calibration with the position method at 20 mm/s.

| | **Quadrant** | $P_r$ **(mm)** | $\theta X_r$ **(°)** | $\theta Y_r$ **(°)** |
|---|---|---|---|---|
| | First | ±0.118 | ±0.147 | ±0.129 |
| **Repeatability** | Second | ±0.130 | ±0.119 | ±0.123 |
| | Third | ±0.192 | ±0.216 | ±0.130 |
| | Fourth | ±0.120 | ±0.140 | ±0.108 |

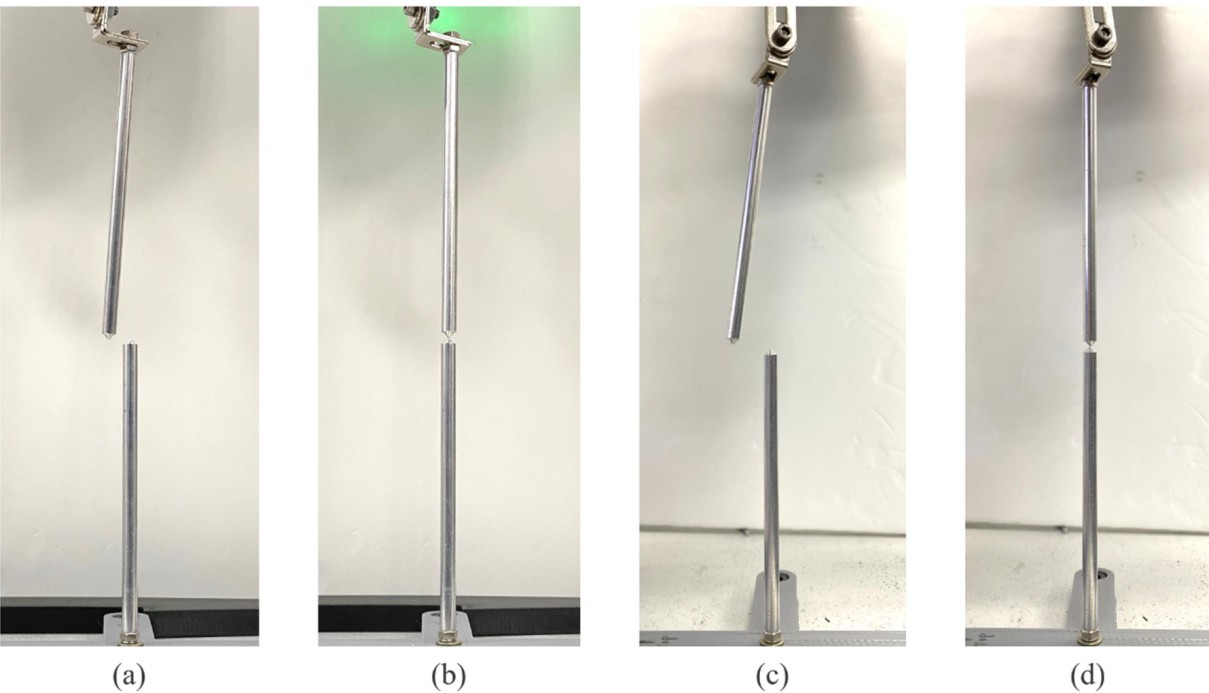

(a)  (b)  (c)  (d)

**Figure 13.** Comparison of before and after calibration using the position method at 20 mm/s speed through the first quadrant calibration experiment: (**a**) *x*-axis before calibration; (**b**) *x*-axis after calibration; (**c**) *y*-axis before calibration; (**d**) *y*-axis after calibration.

*3.5. Results of the with LATC*

For comparing with LATC's existing product, the LATC's calibration software is used to perform the four-quadrant experiment and record the data. First, a calibration speed of 40 mm/s was setup in LATC's software compared with proposed method of 60 mm/s shown in Table 11. Second, the calibration process was almost identical but the motion flow was still a little different. For LATC's entire process, it was programmed in the TM flow system, meaning that all motion, trigger IO, matrix calculations, and position settings were in the same system. For our proposed method, the TM flow system is used to present the robot motion, sensor triggering, and data exchange to the computer, and then use C# to receive the robot information for calculation, processing algorithm, and to send the robot arm position back to the robot. Third, the whole process of using software for LATC is about 70 s, however, our proposed method was about 50 s.

**Table 11.** Comparison of results by the position method and LATC software.

| Quadrant | $P_r$ (mm) | | $\theta X_r$ | | $\theta Y_r$ (°) | |
|---|---|---|---|---|---|---|
| | Position Method | LATC Software | Position Method | LATC Software | Position Method | LATC Software |
| 1st | ±0.118 | ±0.055 | ±0.147 | ±0.102 | ±0.129 | ±0.108 |
| 2nd | ±0.045 | ±0.045 | ±0.130 | ±0.085 | ±0.119 | ±0.124 |
| 3rd | ±0.192 | ±0.074 | ±0.216 | ±0.110 | ±0.130 | ±0.125 |
| 4th | ±0.120 | ±0.088 | ±0.140 | ±0.17 | ±0.108 | ±0.128 |

Results of the four-quadrant calibration experiments by the positioning method achieved positioning errors below 0.12 mm and orientation errors below 0.14° but the results obtained by the LATC software were lower than our proposed experimental results, which can achieve positioning errors below 0.07 mm on average and orientation errors below 0.12° on average. To sum up, compared to LATC's software, the error could be

caused by the loss of data exchange between the TM flow and C#. The calibration cycle of position method is 50 s, which is lower than 70 s of LATC.

Although our proposed method was faster than LATC's experiment, the accuracy was also affected, so in the following adjusted experiments, the speed is set to 40 mm/s, which is the same as LATC's experiment.

From the experimental results shown in Table 12, it can be found that the errors of the two calibration methods are relatively close to each other and the results before adjustment using positioning method are not as accurate as the adjusted positioning method; the reason for this is the speed of the robot's movement. The results of the position method are always inferior to LATC because there is no motion flow to the reference point in their calibration procedure. Meanwhile, when the robot moves, a positioning error is generated and the size of this error depends on the type of robot. Therefore, using LATC's software, the unadjusted experimental results can reach a high accuracy, which decreases when we add an additional motion flow to the program.

**Table 12.** Comparison of results by the position method and LATC software with the same calibration speed of 40 mm/s.

| Quadrant | $P_r$ (mm) | | $\theta X_r$ | | $\theta Y_r$ (°) | |
|---|---|---|---|---|---|---|
| | Position Method | LATC Software | Position Method | LATC Software | Position Method | LATC Software |
| 1st | ±0.114 | ±0.126 | ±0.112 | ±0.106 | ±0.099 | ±0.142 |
| 2nd | ±0.168 | ±0.112 | ±0.187 | ±0.118 | ±0.133 | ±0.127 |
| 3rd | ±0.118 | ±0.065 | ±0.112 | ±0.115 | ±0.138 | ±0.133 |
| 4th | ±0.151 | ±0.129 | ±0.160 | ±0.191 | ±0.074 | ±0.163 |

In summary, the comparison results are more reliable and convincing after fixing the initial conditions, reducing the robot motion speed, and using the same motion flow in both experiments for the calibration experiment.

## 4. Conclusions

In this study, an external laser sensor, which is relatively inexpensive, was used to implement an automatic calibration method for the tool center point of a 6-axis manipulator. After the feasibility analysis and validation of the simulation, it was confirmed that this TCP calibration method can correct the runout and offset error of the tool, moreover, this method is simple, affordable, and automatic. The highlights of this paper are listed as following:

1. The proposed method is a non-contact scheme that uses a laser displacement sensor to handle the TCP calibration procedure;
2. After feasibility analysis and laboratory verification, it is confirmed that this calibration method can calibrate runout, tool offset, and runout plus tool offset error;
3. Although the absolute positioning accuracy of the manipulator is 0.05 mm, the position calibration error is about 0.07 mm to 0.19 mm, and the calibration error of the projection angle is within 0.18°;
4. This automatic tool center point calibration method has the advantages of being simple, versatile, less time-consuming, and relatively inexpensive, and it enables an automatic workflow that maintains the flexibility of the manipulator in the work process.

In conclusion, the TCP calibration method provided in this thesis is an accurate and repeatable one that can be used to improve the pose accuracy and repeatability of industrial robots for point-to-point applications.

**Author Contributions:** Conceptualization, C.-J.L.; methodology, C.-J.L.; software, H.-C.W.; validation, C.-J.L., H.-C.W. and C.-C.W.; formal analysis, C.-J.L. and H.-C.W.; investigation, C.-J.L. and H.-C.W.; writing—original draft preparation, C.-J.L., H.-C.W. and C.-C.W.; writing—review and editing, C.-C.W.; visualization, C.-C.W.; supervision, C.-C.W.; project administration, C.-C.W.; funding acquisition, C.-J.L. All authors have read and agreed to the published version of the manuscript.

**Funding:** This research was funded by the National Science Council of the Republic of China, grant number Contract No. MOST 108-2221-E-027-112-MY3 and MOST111-2622-E-167-019. The APC was funded by Contract No. MOST 108-2221-E-027-112-MY3 and MOST111-2622-E-167-019.

**Institutional Review Board Statement:** Not applicable.

**Data Availability Statement:** Not applicable.

**Conflicts of Interest:** The authors declare no conflict of interest.

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
