# Peer review of "Automatic Calibration of Tool Center Point for Six Degree of Freedom Robot"

_actuators, doi:10.3390/act12030107_

Round 1

Reviewer 1 Report

This paper proposes an enhanced automatic TCP calibration method based on laser displacement sensor. The method is described in detail and the structure is relatively complete, and effective simulation and experimental verification are carried out. Finally, the experimental results are analyzed quantitatively and qualitatively. However, there are still some contents that need to be further improved. The comments are given as follows, which may be helpful to improve the paper quality.

(1) In the introduction of this paper, although the research background is described in detail, the detailed steps and advantages of the proposed method are not well explained.The author should explain the difference from other scholars' research and elaborate on the focus of this work. It is strongly suggested that the references need to make in-depth comments on the content of the cited papers; avoid generic comments. Mention/comment the relevance of the cited paper and especially the research gap associated to it. In addition, there are more relevant papers that should be covered in literature review:

https://doi.org/10.1016/j.tws.2021.107540

https://doi.org/10.1016/j.ymssp.2021.108727

https://doi.org/10.1016/j.ymssp.2022.109225

https://doi.org/10.1016/j.ymssp.2021.107755

https://doi.org/10.1016/j.ymssp.2017.11.046

https://doi.org/10.1016/j.ymssp.2020.106840

(2) On the whole, the quality of the figures involved in this paper is too low, and many of the information in the figures are not specified in detail, which makes it difficult for readers to understand.

(3) Please explain whether the accuracy of the laser displacement sensor involved in the paper will affect the calibration effect.

(4) Please carefully check whether the coordinate system of the last two tool status types (“Runout” and “Offset+Runout”) in Figure 4 (b) is correct.

(5) In Table 3 and Table 4, the length of some experimental data obtained for the experiment is not consistent.

(6) How to determine the moving speed of the robot mentioned in the paper, and whether it can be analyzed and discussed whether the size of the speed will affect the calibration results.

Author Response

Responses :

The authors appreciate reviewers’ valuable comments very much. Modifications have been made to improve the manuscript according to reviewers’ suggestions as summarized below:

  • In the introduction of this paper, although the research background is described in detail, the detailed steps and advantages of the proposed method are not well explained. The author should explain the difference from other scholars' research and elaborate on the focus of this work. It is strongly suggested that the references need to make in-depth comments on the content of the cited papers; avoid generic comments. Mention/comment the relevance of the cited paper and especially the research gap associated to it. In addition, there are more relevant papers that should be covered in literature review:

https://doi.org/10.1016/j.tws.2021.107540

https://doi.org/10.1016/j.ymssp.2021.108727

https://doi.org/10.1016/j.ymssp.2022.109225

https://doi.org/10.1016/j.ymssp.2021.107755

https://doi.org/10.1016/j.ymssp.2017.11.046

https://doi.org/10.1016/j.ymssp.2020.106840

Response: According to the reviewer’s comments, we have enhanced the difference from other scholars' research and elaborated on the focus of this work. Meanwhile, we added more relevant papers in literature review according to the reviewer’s comments. Please refer the revised paper in pages 1-3 and 20.

  • On the whole, the quality of the figures involved in this paper is too low, and many of the information in the figures are not specified in detail, which makes it difficult for readers to understand.

Response:

Thank you very much for your comments. We have enhanced the resolution of figures in our paper. Please refer the revised paper.

  • Please explain whether the accuracy of the laser displacement sensor involved in the paper will affect the calibration effect.

Response:

Thank you very much for your comments. The repeatability and accuracy of laser displacement sensor we used are fulfilled the requirement of the calibration process. The specification of laser displacement sensor (LTC120120) is shown in Table 2 of this paper.

  • Please carefully check whether the coordinate system of the last two tool status types (“Runout” and “Offset+Runout”) in Figure 4 (b) is correct.

Response:

Thank you very much for your comments. We have revised the Figure 4 (b) about “Runout” and “Offset+Runout”. Please refer the revised paper in page 5.

  • In Table 3 and Table 4, the length of some experimental data obtained for the experiment is not consistent.

Response: Thank you very much for your comments. In Table 3 and 4, the Accuracy and Repeatability is the average of the simulation results in Fig. 11. Therefore, the length of Table 3 and 4 is not consistent with the results in Fig. 11.

  • How to determine the moving speed of the robot mentioned in the paper, and whether it can be analyzed and discussed whether the size of the speed will affect the calibration results.

Response: The moving speed of the robot mentioned in the paper can be setup by robot controller shown in Figure 9. Meanwhile, it is found that the accuracy and repeatability under 20mm/s are better than the results of 50 mm/s for the cases of error free stability simulation with the position method. The reason is when using high speed tool through the laser sensor, the sensor can’t be responded immediately and the error will be increased. In addition, for the RoboDK virtual environment, the triggering of the laser sensor is interfered by the 3D model. There is almost no error in the case of speed 20 mm/s or the error is fixed, but the results for speed 50 mm/s are unstable shown in Fig. 11 of this paper.

Reviewer 2 Report

The paper presents an approach for the measurement and calibration of the tool center point (TCP) position for different kinds of tools mounted in a manipulator robot. Apart from other approaches the presented one is based on two-axial laser barrier measurement of the tool respecting the runout error of the tool as well as the tool orientation in space. According to the authors, calibration procedure can be conduct faster than by using other methods.

The paper presents the state of the art in a good matter, while in most parts clearly discussing the advantages and disadvantages of the known approaches as well as their own approach. Nevertheless, the research methodology has some major mistakes, which should be addressed before publication. When these comments are taken into account, the applicability in the industrial approach can be assessed:
- While the authors are presenting the measurement results of their calibration approach, they are not giving information about repetition measurements, or, if they did some, the number of repetitions. The presentation of the repeatability values (Tables 6 and 7) doesn't become clear, as the authors don't explain the methodology to calculate these values. The repeatabilities of 0.0 mm/°/° in table 6 seems as if no repetitions were conducted, so repeatability wouldn't make sense.
- The authors are stating that their approach leads to major measurement deviations when moving the robot at higher speeds (40 mm/s and 50 mm/s), which is reasoned by the time lag for detecting the tool using the laser barrier. Based on the presented results, this statement can not be proven. Another reason for the deviations could be the speed-dependent positioning accuracy of the robot. The authors should discuss this, or, analyze it by doing repetition trials with movements into the laser barrier from both directions, i.e., X+ and X- as well as Y+ and Y-. Also, measurements of the positioning accuracy of the robot at different moving speeds would help to analyze the true error of the robot's positioning accuracy as well as the repeatability of the calibration approach/system. Without these clarifications, the usability of the presented approach for industrial application cannot be assessed.

Minor bullets:
- The calibration methodology is presented well in most parts. The authors should just have another look at the consistency of the variable signs, e.g., for the TCP position, different signs are used throughout the paper.
- Many references to literature, figures and tables are broken ("Error! Reference source not found.").
- The deviation given in line 27 should be discussed in combination with the positioning deviation of the robot.
- Line 76: "to maximize"
- Line 79: "for our sphere fitting algorithm" --> I think the "our" is wrong here?
- Line 86: Capitalize the "I" in "in order to calibrate"
- Line 98: "A physician" may be the wrong word. Please check this.
- Line 102: "magnitude of the robot configuration set" --> Please check if this is correct, maybe you should explain this shortly."
- Line 120: "be calibrated"
- Lines 120-122: Is the calibration really done by human eye, or is this just a quick check?
- Line 139: I guess it should be "relative to the reference coordinate preset" instead of "to the TCP"
- Figure 4: The illustrations of runout and offset+runout are switched
- Line 186: Fullstop at the end of the line is missing.
- Line 189: "orientation" instead of "attitude"?
- Table 2: In the line "Supply Voltage", in the last column a "=" is missing
- Line 415: maybe "depends on the type of robot" instead of "brand of the robot"

Author Response

Response:

The authors appreciate reviewers’ valuable comments very much.

  • - While the authors are presenting the measurement results of their calibration approach, they are not giving information about repetition measurements, or, if they did some, the number of repetitions. The presentation of the repeatability values (Tables 6 and 7) doesn't become clear, as the authors don't explain the methodology to calculate these values. The repeatabilities of 0.0 mm/°/° in table 6 seems as if no repetitions were conducted, so repeatability wouldn't make sense.

Response: In section “3.3 Experiment of Error-Free Stability”, according to the actual error-free stability experiment, standard of calibration tools with a processing error of ±0.05mm are used, and the whole process is performed 30 times consecutively. After calibration, the tool is moved to the observation station to check the result. The moving speed of robot is set to be 20mm/s and 40mm/s, respectively.

In section “3.4 Four-Quadrant Calibration Experiment”, the tool was used in the same way as the error-free experiment. Before the experiment starts, the tool is rotated to the four-quadrant position for the experiment, and then the calibration procedure is started and calibrated 30 times continuously, and after the calibration is finished, the tool is moved to the observation station and the results are checked. Due to the offset of the rotating tool, the actual TCP and tool position are difficult to measure, so the stability results of the calibration are only considered. The results of the first quadrant calibration experiment with the position method at a speed of 20 mm/s are shown in Fig. 12, and the experimental results have been compiled in Table 10.

For the repeatability, we follow the ISO9283 to calculate the positioning accuracy, orientation accuracy, positioning repeatability and orientation repeatability, and the definition of positioning accuracy and positioning repeatability shown in Figure A1.

Figure A1 Positioning accuracy and repeatability[R1]

After an accuracy test with n test points, the point  represents the command position, whose coordinates are , and the origin of the reference coordinate system, and X, Y, and Z represent the three axes of the reference coordinate system, and the coordinates of random test point are , while the point  is the average of all test points in the accuracy test with coordinates . The distance between the point and the origin is the absolute positional accuracy, which is given as , The spherical radius of the smallest sphere formed by the centers of all test points is the repeatability accuracy, which is given as , and all the related accuracy and error equations are shown in Equations 1 to 11.

                                                                                         (1)

                                                                                         (2)

                                                                                          (3)

                                                                                       (4)

                                                                                        5)

                                                                                       (6)

                                    (7)

                                         (8)

                                                                                           (9)

                                                                                 (10)

                                                                                     (11)

        In Equation 8 to 11, where  is the distance between test point and point ,  is the average value of , and  is the standard variation of .

The Figure A2 shows the orientation accuracy and repeatability.

Figure A2 Orientation accuracy and repeatability [R1]

In Figure A2, we can know that after an accuracy test of n test points, the rightmost intersection point is the visual projection point of the random test point in one of the three axes X, Y and Z in the reference coordinate system, the three projection angles in the runout calibration space are a, b and c. There is no specific axis, and the projection angles of each test point pose are given as , , , respectively.  is the average projection angle of all test points,  is the error between  and , which is the absolute orientation accuracy of the projection angle c, and  is the standard variation of , and finally  is the repeat orientation accuracy of the projection angle c. All the related accuracy and error Equations are shown in Equations 12 to 20.

                                                                                       (12)

                                                                                       (13)

                                                                                        (14)

                                                                                     (15)

                                                                                     (16)

                                                                                      (17)

                                                                       (18)

                                                                       (19)

                                                                       (20)

The above accuracy and repeatability equations are used to measure, calibrate and validate the robot arm, however, there is no particular technical validation for TCP calibration, however, the accuracy and repeatability equations provided by ISO 9283 [R1] are quite commonly used in mechanical and automation technology, so this study used the calibration error calculation and calibration stability equations for subsequent experiments with reference to the calibration error or calibration stability results.

Reference R1: International Standards Organization, ISO 9283, “Manipulating industrial robots – Performance criteria and related test methods”, 1998.04.01, second edition.

  • The authors are stating that their approach leads to major measurement deviations when moving the robot at higher speeds (40 mm/s and 50 mm/s), which is reasoned by the time lag for detecting the tool using the laser barrier. Based on the presented results, this statement can not be proven. Another reason for the deviations could be the speed-dependent positioning accuracy of the robot. The authors should discuss this, or, analyze it by doing repetition trials with movements into the laser barrier from both directions, i.e., X+ and X- as well as Y+ and Y-. Also, measurements of the positioning accuracy of the robot at different moving speeds would help to analyze the true error of the robot's positioning accuracy as well as the repeatability of the calibration approach/system. Without these clarifications, the usability of the presented approach for industrial application cannot be assessed.

Response: Thank you for your suggestion. Because our experiments need to perform and compare at higher speeds (40 mm/s and 50 mm/s), therefore we only consider the positioning accuracy of the robot at 40mm/s and 50 mm/s. Indeed, measurements of the positioning accuracy of robot depends on the moving speeds. With other speeds of the tasks, we need to perform our experiments again.

  • The calibration methodology is presented well in most parts. The authors should just have another look at the consistency of the variable signs, e.g., for the TCP position, different signs are used throughout the paper.

Response: Thank you for your suggestion. Due to calculation of the TCP position is applied in different cases, different signs are used. For example, offset calibration, Fig. 7 shows that O, O' and O'' are the ideal tool coordinate, runout tool coordinate and runout with offset tool coordinate system, respectively. Sc (Xs, Ys, Zs) is the tool installed station, and Pc (X0,Y0,Z0), Pc'(X0',Y0',Z0') and Pc''(X0'',Y0'',Z0'') are the original points of the ideal tool coordinate, runout tool coordinate and runout with offset tool coordinate system, respectively.

Figure 7. Coordinate space model.

  • Many references to literature, figures and tables are broken ("Error! Reference source not found.").

Response: Thank you for the reviewer’s comments and we have modified the hyperlinks in the Reference. Please refer the revised paper in page 20.

  • The deviation given in line 27 should be discussed in combination with the positioning deviation of the robot.

Response: In line 27, we have modified the discussion in combination with the positioning deviation of the robot and please refer the revised paper in pages 1, 14, and 16.

  • - Line 76: "to maximize"

- Line 79: "for our sphere fitting algorithm" --> I think the "our" is wrong here?

- Line 86: Capitalize the "I" in "in order to calibrate"

- Line 98: "A physician" may be the wrong word. Please check this.

- Line 102: "magnitude of the robot configuration set" --> Please check if this is correct, maybe you should explain this shortly."

- Line 120: "be calibrated"

- Lines 120-122: Is the calibration really done by human eye, or is this just a quick check?

- Line 139: I guess it should be "relative to the reference coordinate preset" instead of "to the TCP"

- Figure 4: The illustrations of runout and offset+runout are switched

- Line 186: Fullstop at the end of the line is missing.

- Line 189: "orientation" instead of "attitude"?

- Table 2: In the line "Supply Voltage", in the last column a "=" is missing

- Line 415: maybe "depends on the type of robot" instead of "brand of the robot"

Response: Thank you very much for your detail comments. We have checked and revised for all of the problem above and please refer the revised paper.

Round 2

Reviewer 2 Report

Dear authors,

thank you very much for your clarifications in the paper, which clearly help the understanding.
Still, there is one bullet you should think about regarding the precision of the newly developed measurement strategy for robot TCP calibration (please also refer to my second bullet in the first review round).
The position deviation of the robot is generated by, e.g., the drives, control system, position measurement system and compliance of the robots structure and inertia. When comparing the position measurements of the robots integrated position measurement systems with the laser barrier detection, the main deviation will be generated by the robot, as the laser barrier is detecting the tool or calibration pin in light speed. Thus, in my opinion, the calibration accuracy of the presented system could be highly improved by measuring twice, moving to the laser barrier from both directions. This procedure could work regardless of the moving speed of the robot.
Please think about conducting these measurements again to improve the paper. You don't need to conduct the measurements at different moving speeds, but an assessment at one fixed speed, i.e., 40 or 50 mm/min, would be sufficient.
Otherwise, I would recommend at least to add a short discussion about this error source and simple method for improving your results to the Outlook section.

Some small bullet points:
- In lines 60 and 85, opening square brackets for the references are missing.
- In line 137, "the TCP" should be deleted.

Kind regards.

Author Response

Response:

The authors appreciate reviewers’ valuable comments very much.

  • - Still, there is one bullet you should think about regarding the precision of the newly developed measurement strategy for robot TCP calibration (please also refer to my second bullet in the first review round).

The position deviation of the robot is generated by, e.g., the drives, control system, position measurement system and compliance of the robots structure and inertia. When comparing the position measurements of the robots integrated position measurement systems with the laser barrier detection, the main deviation will be generated by the robot, as the laser barrier is detecting the tool or calibration pin in light speed. Thus, in my opinion, the calibration accuracy of the presented system could be highly improved by measuring twice, moving to the laser barrier from both directions. This procedure could work regardless of the moving speed of the robot.

Please think about conducting these measurements again to improve the paper. You don't need to conduct the measurements at different moving speeds, but an assessment at one fixed speed, i.e., 40 or 50 mm/min, would be sufficient.

Otherwise, I would recommend at least to add a short discussion about this error source and simple method for improving your results to the Outlook section.

Response: Thank you the comments and reminds from reviewer.

In Figure 8 of the revised paper, the first preliminary offset calibration is performed after the 2nd plane movement is completed. The first preliminary offset calibration, which estimates the tool height Zh, can be performed when the tool only has an offset. The second preliminary offset calibration is performed by deriving the tool offsets δX and δY from the tool center offset equation. Figure R1 represents how the tool height Zh is calculated.

Figure R1. The tool height derivation chart of the tool runout coordinate system

Looking in from the positive direction of the Y-axis of the coordinates of the tool mounting point, where O is the tool mounting station and the origin of the coordinate system, the coordinates of are (0,0,Zh) and the coordinates of  are (x',y',z'). Let the first plane of calibrated motion be M1 and the second plane be M2, where  intersects with plane M1 at  and with plane M2 at ,  intersects the plane M1 at  and   intersects the plane M2 at , and the parameters ∆x1, ∆x2, ∆y1, ∆y1 have been obtained from Equation 6. Let the height from plane one to , labeled  be ∆H, and the height between plane one and plane two, labeled be ∆h. After defining the above information, the following Equations (Re1) and (Re2) can be obtained.

    Zh =Z1+ ∆H    (Re1)

    Z2 =Z1+ ∆h    (Re2)

Where Zh is the tool height that needs to be derived, ∆H and ∆h is the parameter set by the user. Use space linear proportion and the center offset parameter, which is ∆x1,  ∆x2,  ∆y1,  ∆y1, Then, through relational substitution can solve the Z1and Z2.

To express the spatial linear scale, the coordinates of at least two points in space must be known, herby, assume there are two points A(x1,y1,z1) and B(x2,y2,z2), the line η between these two points can be expressed in the spatial linear scale as Equation (Re3)

        (Re3)

    Similarly, since ∆x1,  ∆x2,  ∆y1,  ∆y1, z1, z2 can construct two points 〖  and , the linear Η is expressed by the spatial linear scale,

           (Re4)

   In this case, since the line Η passes through the tool mounting point, after substituting Equation (Re4) and combining Equation (Re2) with Equation (Re1), then we can obtain Zh as Equation (Re5).

        (Re5)

   After the above calculation, it seems that the tool height Zh has been solved, however, a tool with offset error will result in a denominator equal to zero, while a tool with offset and offset error will result in a line Η not passing through the tool mounting point O. Therefore, for runout error or runout plus offset error, the original tool height will be used directly as the tool length Zh.

   In the first preliminary offset calibration only roughly calculated the tool height, runout plus offset error of the coordinates still can not be solved, so in the first preliminary offset calibration, just Zh generation back to Equation (11) in the revised paper, to get the tool center point that is the current tool coordinate origin, set to , and the coordinates will be returned to the robot arm controller, the first preliminary offset calibration is completed.

            Through the first preliminary offset calibration process, the position deviation of the robot is can be calculated precisely and also to reveal the all the parameters in the calibration process. We also added a short discussion about the error source and simple method for the calibration method in the revised paper. Please refer to the revised paper in pages 11-12.

  • Some small bullet points:

- In lines 60 and 85, opening square brackets for the references are missing.

- In line 137, "the TCP" should be deleted.

Response: Thank you very much for your detail comments. We have checked and revised for all of the problem above and please refer the revised paper.
